# Highly Transparent Red Organic Light-Emitting Diodes with AZO/Ag/AZO Multilayer Electrode

**DOI:** 10.3390/mi15010146

**Published:** 2024-01-18

**Authors:** Dongwoon Lee, Min Seok Song, Yong Hyeok Seo, Won Woo Lee, Young Woo Kim, Minseong Park, Ye Ji Shin, Sang Jik Kwon, Yongmin Jeon, Eou-Sik Cho

**Affiliations:** 1Department of Electronics Engineering, Gachon University, Seongnam 13120, Republic of Koreasjkwon@gachon.ac.kr (S.J.K.); 2Department of Biomedical Engineering, Gachon University, Seongnam 13120, Republic of Korea

**Keywords:** TCO, transparent electrode, AZO/Ag/AZO, transparent OLED

## Abstract

Free-form factor optoelectronics is becoming more important for various applications. Specifically, flexible and transparent optoelectronics offers the potential to be adopted in wearable devices in displays, solar cells, or biomedical applications. However, current transparent electrodes are limited in conductivity and flexibility. This study aims to address these challenges and explore potential solutions. For the next-generation transparent conductive electrode, Al-doped zinc oxide (AZO) and silver (AZO/Ag/AZO) deposited by in-line magnetron sputtering without thermal treatment was investigated, and this transparent electrode was used as a transparent organic light-emitting diode (OLED) anode to maximize the transparency characteristics. The experiment and simulation involved adjusting the thickness of Ag and AZO and OLED structure to enhance the transmittance and device performance. The AZO/Ag/AZO with Ag of 12 nm and AZO of 32 nm thickness achieved the results of the highest figure of merit (FOM) (Φ_550_ = 4.65 mΩ^−1^) and lowest roughness. The full structure of transparent OLED (TrOLED) with AZO/Ag/AZO anode and Mg:Ag cathode reached 64.84% transmittance at 550 nm, and 300 cd/m^2^ at about 4 V. The results demonstrate the feasibility of adopting flexible substrates, such as PET, without the need for thermal treatment. This research provides valuable insights into the development of transparent and flexible electronic devices.

## 1. Introduction

Recently, in the era of the 4th Industrial Revolution, the significance of hyperconnectivity is emphasized [1]. The Internet of things (IoT), metaverse, fifth-generation (5G) wireless networks, artificial intelligence (AI) [2], virtual reality (VR), mixed reality (MR), augmented reality (AR), and electric vehicles are examples [3]. For this reason, the importance of electronics with form factors that allow free attachment to the human body is increasing. Free-form factors mean that the device should be flexible, bendable, or stretchable to be customized to fit any individual’s body [4]. Among these factors, research is essential to advance the development of displays that can efficiently convey information to individuals. But at present, limitations of space, objects, and people exist. For example, flexible displays are only commercialized for TVs and foldable phones.

In particular, optoelectronics grounded in light offers scope for substantial ongoing research in semiconductors, displays, sensors, and biomedical applications. Given the light-based nature of optoelectronics, the transparent electrode is of notable significance. This is becoming increasingly evident in the development of next-generation displays, such as light-emitting diodes (LEDs) and OLEDs, among others [5].

Also, to reach free-form factor application, features like flexibility are becoming increasingly crucial for emerging applications, such as wearable devices or implantable devices that can be integrated into the human body. Examples include wearable OLEDs for disposable skin wound photomedicine [6], wearable OLED photo biomodulation patch in vitro differential using cell proliferation effects [7], wearable blue OLED photomedicine for neonatal jaundice treatment [8], real-time color changing of OLEDs synchronized with the shape of the measured (top) normal electrocardiography (ECG) signal and (bottom) abnormal ECG signal, optoelectronic skins for optical visualization and tactile sensing, and wireless communicating systems with integrations of pressure sensor arrays and tactile-interactive OLED [9].

Although many transparent electrodes exist, such as thin metal film, grid, nano particles, nano wires, transparent conducting oxide (TCO), and graphene [9], it is not easy to achieve transparent and flexible properties, so research continues. This is because the most frequently utilized electrode in current displays is indium tin oxide (ITO) [5]; however, ITO exhibits limited flexibility. It also has the limitation of ITO employing indium, a scarce metal, making it costly and toxic. In industry, large displays require a sheet resistance of only a few Ω/Υ. Achieving this requires heat treatment of the indium tin oxide (ITO), which involves high temperatures (>300 °C) that plastic flexible substrates cannot withstand [10]. To address this issue, numerous research efforts have been made, and to overcome these challenges, a method involves depositing a metal with high transmittance and low sheet resistance, such as silver (Ag), between the layers of ITO. This results in a structure known as dielectric/metal/dielectric (D/M/D) or TCO/Metal/TCO (T/M/T).

In this study, we aim to address the existing limitations and challenges specifically in the transmittance for AR, MR, and electronic vehicles, as well as photomedicine. A similar TCO material to ITO, aluminum-doped zinc oxide (ZnO:Al or AZO), is a promising alternative to replace ITO and offers the advantages of high transmittance. Compared to ITO, it offers low crystallization temperature of less than 200 °C, low cost, and abundant resources [11]. The disadvantages of AZO of exhibiting poor sheet resistance compared to ITO can be overcome by employing the T/M/T structure. Metals such as Ag, which have high transmittance in the visible region, also become opaque as they become thicker. Therefore, it is important to increase the transmittance by appropriately adjusting the thickness of the AZO and Ag. This adjusted electrode was applied to a transparent OLED (TrOLED) device to confirm that it could be used as a transparent electrode. A transparent organic light-emitting diode (OLED) display is a highly promising device for the next-generation displays, including AR/VR/MR, smart glass, and smart windows, due to its advantageous characteristics of flexibility [12,13,14] and self-emissive properties with optical clarity.

We successfully applied the optimized AZO/Ag/AZO electrodes to red TrOLEDs with high-transmittance OLED via structural optimization. The electrodes and TrOLEDs are flexible, so the possibility of applying a flexible substrate was also demonstrated. Of course, this electrode could be developed in the future for a next-generation display that is not limited to OLED.

## 2. Materials and Methods

The AZO/Ag/AZO samples were prepared in an in-line magnetron sputtering system. Before the sputtering process, the substrate glass was cleaned with acetone, isopropyl alcohol (IPA), and deionized (DI) water in an ultrasonic bath for 10 min. After the cleaned glass was vertically loaded into a jig in the load-lock chamber, it was moved to the process chamber of the in-line sputter. Figure 1a shows that, for the first layer, the glass substrate was moved in front of a 99.99% AZO (2 wt.% Al_2_O_3_-doped ZnO) target of 165 mm × 540 mm × 7 mm in size at a base vacuum of 8.5 × 10^−6^ Torr, and plasma was generated with a magnetron pulse DC power of 2 kW with 200 kHz pulse frequency and 0.5 µs pulse reverse time, at a pressure of 5 mTorr and at room temperature (RT). Ar gas injections were maintained at 50 sccm. During the AZO sputtering, the jig was scanned in front of the AZO target at a speed of 60 cm/min (20 Hz), to reach a thickness of 32 nm.

Figure 1a shows that for the inter-layer, the glass substrate was moved in front of a 99.99% Ag target of 4 in diameter and 1/8 in thickness at RT. Plasma was generated using magnetron radio frequency (RF) sputtering with an RF power of 60 W at a pressure of 3 mTorr. Ar gas injections were maintained at 30 sccm. During the Ag sputtering, the jig was scanned in front of the Ag target at a speed of 150 cm/min (50 Hz), to reach a thickness of 6 nm. At the 12 nm Ag condition, the jig was scanned twice. For the third AZO layer, the same method as described above was conducted. All processes were performed without breaking the vacuum. For the anode patterning, a shadow mask was used in the sputtering process.

For the OLED device process, all OLED layers were deposited via a thermal evaporation method in the vacuum chamber, without breaking the vacuum. The chamber atmosphere was 1.7 × 10^−7^ torr, which was obtained using a cryo pump. MoO_3_ (deposition rate: 0.5 Å/s; thickness: 5 nm) was used as the hole injection layer (HIL), NPB (deposition rate: 1 Å/s; thickness: 40 nm) was used as HTL, Bebq_2_ (deposition rate: 0.92 Å/s, 41 nm) was used as the EML host, and Ir(piq)_3_ (deposit rate: 0.08 Å/s, 9 nm) was used as the EML dopant, while Cs_2_CO_3_ (0.1 Å/s, 1 nm) was used as the EIL. A Mg:Ag cathode was deposited at a 1:9 ratio with 0.1:0.9 Å/s. MoO_3_ (deposit rate: 0.5 Å/s; thickness: 45 nm) was used as the CPL. The sample was encapsulated in glass, sealed with UV epoxy resin.

The sheet resistance was measured using a 4-point probe method. UV–Vis spectrophotometry (Lamda-35, PerkinElmer, Waltham, MA, USA) was used to measure the transmittance. A spectroscopic radiometer (CS2000, Konica Minolta, Chiyoda City, Tokyo) and a source meter (2401, Keithley, Cleveland, OH, USA) were used to measure the opto-electrical characteristics of the SETOLEDs. Magnification images were obtained by SEM (SU8600, Hitachi, Chiyoda City, Tokyo). Surface roughness was measured by AFM (NX10, Park, Suwon, Republic of Korea). The simulation was conducted with a custom-made MATLAB R2022 code.

## 3. Results and Discussion

### 3.1. Optimization of the Electro-Optical Properties of the AZO/Ag/AZO Transparent Electrodes

Figure 1a shows how to optimize the AZO/Ag/AZO thin film. AZO/Ag/AZO was deposited in-line, without breaking the vacuum, and thermal treatment. Depositing the first layer of AZO, Ag was deposited in the arrow 1 direction, and then returned to the AZO target, as indicated by arrow 2. The third AZO layer covered the Ag under the same conditions as the first AZO layer.

Figure 1b shows the simulation performed to optimize the transmittance of the thin film. The first layer of AZO and the third layer of AZO were set to the same thickness, so the thickness in the simulation was equivalent to being stacked twice.

To ensure consistent thin film characteristics, it was utilized as a bottom or top electrode. The sandwiched second layer of Ag was simulated limited to down to 20 nm to allow transparency, lower the sheet resistance, and improve flexibility. The simulation results showed that as the AZO thickness increased, the average transmittance initially increased about 42 nm, and then decreased.

While the simulated average transmittance may be higher in the visible light range, the peak transmittance moves away at 555 nm, which is considered to be the most transparent wavelength, based on the luminosity factor. Therefore, the AZO thickness in Figure 1c was not increased beyond this point. As the Ag became thicker, the transmittance decreased, depending on the thickness of Ag. Consequently, to determine the transmittance graph depending on the wavelength, Ag thickness of 12 nm was chosen and simulated starting from approximately 42 nm thickness of AZO, decreasing by 10 nm. The crucial factors, the conductivity, and the transmittance of such T/M/T structures are mainly determined by the thickness of the metal layer [15]. It is well known that continuous film of Ag is not easy to form, due to the island formation phenomenon, termed Volmer–Weber growth, resulting in high resistance and high roughness [16] to form a film, typically at a standard thickness of (5–7) nm or higher [17,18,19]. So, we selected thicknesses of (6 and 12) nm, which could be formed into film. We then examined the transmittance, focusing on the 12 nm thickness, which is presumed to have the lowest transmittance.

As shown in Figure 1c, to verify the enhancement in transmittance, a single layer of Ag with a thickness of 12 nm was also employed for comparison, and demonstrated the lowest transmittance. While the thickness of Ag layers was fixed at 12 nm, the AZO/Ag/AZO with an AZO single layer thickness of 32 nm appeared to have the highest transmittance in the visible light region. However, as the AZO layer increased or decreased in thickness, the transmittance decreased. Therefore, the optimal AZO condition was set at 32 nm. To improve the figure of merit (FOM) of the transparent conductive electrodes (TCEs) of the multilayer structure of AZO/Ag/AZO, the thickness of Ag was set to (0, 6, and 12) nm, while keeping the AZO thickness constant at 32 nm.

In the case of transparent electrodes, the electric current typically depends on the thickness of the electrode. The electrical conductivity and optical transmittance exhibit a conflicting relationship. Therefore, to compare the characteristics of transparent electrodes, various performance indices, such as the figure of merit (FOM), have been proposed. The performance index defined by Haake, the figure of merit $\Phi_{TC}$ (FOM), was calculated using Equation (1) [20]:(1)$$\Phi_{TC}=T^{10}/R_{S}$$
where *T* is the optical transmission and *R_s_* is the sheet resistance. The transmittance and sheet resistance to calculate the FOM were measured through actual deposition. Figure 1d shows the measured transmittances of the AZO/Ag/AZO multilayers and Ag single layers. The AZO double layer without Ag showed the highest transmittance (82.77% at 550 nm), while the 6 nm Ag condition (77.06% at 550 nm) showed slightly better transmittance than the 12 nm Ag condition (76.62% at 550 nm). This is because optical resonance caused by two AZO film layers allows for higher transmittance than pure Ag film [21]. Figure 1e shows the measured sheet resistance $R_{S}$ of the AZO/Ag/AZO multilayers and Ag single layers. The single layer of the 12 nm Ag condition (12.49 Ω/Υ) was the lowest condition, while the AZO/Ag/AZO multilayer with 12 nm Ag shows similar sheet resistance (12.22 Ω/Υ). The AZO/Ag/AZO multilayer with 6 nm Ag shows a sheet resistance of 64.17 Ω/Υ, while the AZO double layer shows the highest sheet resistance of 717.03 Ω/Υ.

Based on these results, Figure 1f shows that the AZO/Ag/AZO multilayer with a Ag thickness of 12 nm has the highest FOM (Φ_550_ = 4.65 mΩ^−1^), followed by the AZO/Ag/AZO multilayer with a Ag thickness of 6 nm (Φ_550_ = 1.15 mΩ^−1^), AZO double layer (Φ_550_ = 0.21 mΩ^−1^), and the lowest single Ag layer (Φ_550_ = 0.17 mΩ^−1^). These results reveal that the T/M/T AZO/Ag/AZO multi-layer possesses superior characteristics, compared to the single AZO or Ag material.

OLED consists of a very thin film (scale of hundreds of nm), and if the electrode has high roughness, there is a risk of short-circuiting between the anode and cathode. Consequently, it needs low surface roughness and proper thin film formation [22]. We evaluated the transparent conductive electrodes (TCEs) through transparent OLED devices, by analyzing the surface roughness via SEM and AFM, a crucial factor when using it as an OLED anode. Figure 2 shows the SEM image and AFM R_pv_, R_q_, and R_a_ values, while Figure 2a–c show the AZO/Ag/AZO with a Ag thickness of (0, 6, and 12) nm. In the scanning electron microscopy (SEM) images under all conditions, the grain size of AZO appears sufficiently flat, indicating that there seems to be no issue for OLED deposition. As the Ag inter-layer thickness is increased, all AFM R_pv_, R_q_, and R_a_ values lower, and improve. This improvement is attributed to the fact that when Ag is deposited between AZO layers, the roughness of the AZO layer is smoothed out through the Ag inter-layer as it becomes flattened. Direct stacking of the AZO thin film on top of another AZO layer that is relatively rough can have a negative impact on the surface. The AZO/Ag/AZO structure with a Ag thickness of 12 nm is considered optimal due to it having the highest FOM and lowest roughness, including well-filmed thin films. With these results, the surface was adequately flat, and its feasibility was demonstrated by evaluating the electrode’s performance by depositing TrOLED.

### 3.2. Design of a Transparent OLED Structure Based on the AZO–Ag–AZO

To obtain more transparency, optimization of the TrOLED’s structure is also required. Figure 3a shows the red OLED with the AZO anode. The work function of AZO is sufficiently high to be employed in the OLED anode, and the work function of the AZO thin film is known to be 4.75 eV [23]. This property was further enhanced by a MoO_3_ HIL layer. In TrOLEDs, the transparency of both electrodes is crucial. For use as an OLED cathode, the work function of AZO is too high. Instead, we employed Mg:Ag electrodes, which are more transparent than the Ag electrodes, have a low work function that is suitable for a cathode, and minimize the organic layer’s damage by thermal evaporation. The work function of the Mg:Ag cathode is known to be 4.12 eV [24]. Moreover, an EIL layer is also applied to sufficiently reduce the work function. Instead of blue or green, a red EML was used to make it easier to compare through the transmittance dependency, which shows the most significant wavelength changes depending on the thickness variation of Ag in the AZO/Ag/AZO anode. Appendix A compares FOMs of 630 nm for the emission peak of the red EML.

Additionally, as simulated in Figure 3b, CPL on the cathode was employed to optimize the optical path, and the hole transport layer (HTL) thickness was also adjusted to modify the total OLED layer thickness for transparency.

The EML was fixed at 50 nm, and the HIL was maintained at 5 nm, while the thickness of the hole transport layer (HTL) varied. A thinner HTL thickness can lead to low driving voltage, but it may also result in shorter OLED lifetime, due to the presence of undesired defects and pinholes [22]. Hence, an HTL thickness of 40 nm was chosen, representing a relatively thick value. This thickness corresponds to the red area indicating a high transmittance region in the simulation. The overall device transmittance was then precisely optimized by varying the CPL thickness as (35, 45, and 55) nm, while simulating the transmittance depending on the wavelength (Figure 3b). As shown in Figure 3c–e, full TrOLED structure transmittances were simulated with an AZO/Ag/AZO anode thickness of (0, 6, and 12) nm Ag and an HTL thickness of 40 nm. In all conditions of Ag thickness for the AZO/Ag/AZO anode, it showed more transparency than that without CPL; as shown in Table 1, all conditions of Ag thickness for the AZO/Ag/AZO anode show the highest average transmittance in visible length when the CPL thickness of 45 nm is applied. Consequently, after choosing the OLED’s HTL at 40 nm and CPL at 45 nm, the EL spectra of the TrOLED, a crucial factor affecting color coordinates and efficiency, were simulated. The structure was finally confirmed to ensure whether the emission peak was shifted on both sides, affected by the Fabry–Perot oscillator generated by the two reflective electrodes on both sides [25]. Figure 3f shows that as the Ag inter-layer becomes thicker, weak microcavities occur. However, the spectrum exhibits nearly the same characteristics at the top and bottom. Therefore, the thickness between the two reflective electrodes was fixed as described above. With this structure, Figure 3g shows full surface deposition samples to measure the transmittance, while Appendix A shows the actual photos of the samples. The transmittance in all conditions seemed to be similar. Also, as shown in Table 1, the full device with the AZO/Ag/AZO anode varied with the Ag inter-layer, the average transmittance in the visible range being higher in the 0 nm Ag condition, followed by 12 nm Ag, and then 6 nm Ag. At a wavelength of 550 nm, the 12 nm Ag condition (64.84%) showed the highest transmittance, followed by 0 nm Ag (63.41%), and then 6 nm Ag (58.48%). The transmittance of 12 nm Mg:Ag deposited at a 1:9 ratio was 63.69% [26]. This shows that our TrOLEDs with the AZO/Ag/AZO anode and Mg:Ag cathode offer quite good transmittance, with both sides having metal electrodes [27]. Through this process, a device with optimized transmittance was completed, and its performance was analyzed.

### 3.3. Optimization and Analysis of the AZO–Ag–AZO-Based Transparent OLED

Figure 4a summarizes the final optimized TrOLED structure. Three AZO/Ag/AZO devices with varied thickness of the Ag inter-layer of (0, 6, and 12) nm were fabricated. The device was deposited simultaneously on substrates of three different thicknesses of Ag. The substrates were rotated at the deposition position to ensure uniform thickness deposition under all substrate conditions. As simulated, NPB of HTL is deposited at 40 nm. The red Bebq_2_-doped Ir(piq)_3_ 8 wt.% EML is also used as the ETL, because Bebq_2_ has a sufficient energy level to use the ETL. The thickness of 12 nm Mg:Ag is used, which offers sufficient sheet resistance for the cathode [28]. Finally, the MoO_3_ CPL is deposited in the optimum thickness condition, as simulated. As shown in Figure 4b, the sum of the bottom emission and top emission luminance is highest in the 12 nm Ag condition, followed by 6 nm Ag, and then 0 nm Ag. Figure 4b,c and Appendix A show the photos of turn on and off, which confirm that both the bottom and top sides turned on well. But unlike the conditions with Ag thicknesses of (6 and 12) nm, under the AZO double anode conditions where Ag was not deposited, lights were not observed in specific spots, which are termed dark spots, as shown in the magnified picture of ON in Figure 4c. This phenomenon seems to be caused by a significant peak valley in the rough AZO double layer, making pinholes that cause it to not emit, which is not shown in the small area in the AFM [29]. In the surface geometry model of the AZO/Ag/AZO, the Ag inter-layer deposited between them flattens the initially rough AZO. However, in the AZO double layer, flattening due to a Ag inter-layer is not achievable [30]. The current density was compared according to each Ag thickness, as shown in Figure 4d. Like the luminance characteristics, the current density is highest in the 12 nm Ag condition, followed by 6 nm Ag, and then 0 nm Ag. The reason for the poor current flow of the 0 nm Ag condition is the high sheet resistance, which also lowers the luminance. In the 6 nm Ag condition, the sheet resistance is also high, while the current density is low, so the driving voltage is high. Only the 12 nm Ag condition shows the driving voltage of the ordinary OLED at 4 V. Figure 4e compares the top and bottom total power efficiency. Excluding the initial low luminance values, the 12 nm Ag condition shows equal or higher efficiency than the 6 nm Ag condition. The 0 nm Ag condition shows the lowest power efficiency.

Appendix A shows the total luminance and current density to 40 V, while Appendix A show the top and bottom luminance, current density, and power efficiency. In the case of the 12 nm Ag conditions, the driving voltage was low enough, and both the top and bottom needed to be measured; however, measuring the top side first at a high voltage could potentially damage the device, so the measurement was not conducted beyond 7 V. Therefore, it would have been possible to achieve higher brightness at a higher voltage. Even when separating the top and bottom emissions, the 12 nm Ag condition shows the best characteristics in terms of brightness and current density, compared with the (6 and 0) nm Ag conditions. Finally, to confirm that the EL spectrum is not distorted, the EL spectra for each Ag condition were compared. As shown in Figure 4f, all EL spectra show a similar shape, and the same peak at 628 nm. These indicate that the weak micro-cavity effect did not disturb the comparison of the TrOELD characteristics. If the EL spectra had been different, comparing various conditions would have been challenging, due to the luminosity factor and microcavity effects depending on the organic layer thickness. We successfully implemented the AZO/Ag/AZO TCE for the TrOLED anode.

## 4. Conclusions

In conclusion, the AZO/Ag/AZO multilayers are optimized by the simulation thickness of AZO at 32 nm, with the thickness of the Ag inter-layer that is sandwiched between the two AZO layers varied at (0, 6, and 12) nm. The split conditions are deposited by magnetron sputtering to determine the FOM of the TCE. The AZO/Ag/AZO multilayer with a Ag thickness of 12 nm shows the highest FOM (Φ_550_ = 4.65 mΩ^−1^), followed by a Ag thickness of (6 and 0) nm. The surface roughness was also lowest at a Ag thickness of 12 nm, followed by that of (6 and 0) nm. The transparent OLED was then deposited to use the AZO/Ag/AZO as an electrode. The OLED structures are optimized for transparency. With this structure, the successful implementation of the TrOLED is shown, reaching 64.84% transmittance at 550 nm with the AZO/Ag/AZO multilayer with a Ag thickness of 12 nm. This device also shows 300 cd/m^2^ at about 4 V.

To the best of our knowledge, this work is the world’s first on TrOLED, and it is suitable for applications such as AR, smart glasses, and MR, with the AZO/Ag/AZO electrode that is capable of maximizing the transparency and flexibility for the wearable device. Furthermore, compared to other studies where the AZO or AZO/Ag/AZO anode is used for OLED, despite being a TrOLED, the characteristics of our device are impressive [31,32,33,34,35,36,37,38,39].

The successful result without thermal treatment for the AZO/Ag/AZO TCE and TrOLED also indicates the possibility of adopting flexible substrates with low thermal stability, such as PET [12,37,40,41,42]. This means that the full device can be bent, folded, or rolled for the next-generation display, and we hope this work will not be limited to OLED’s electrode.

## Figures and Tables

**Figure 1 micromachines-15-00146-f001:**
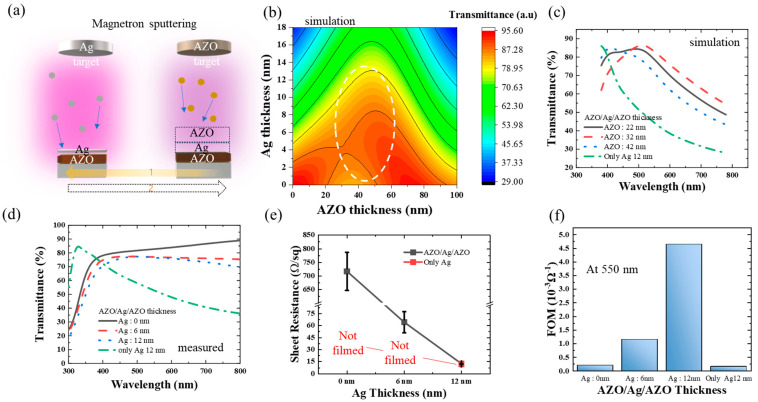
Concept of the optimized electrical and optical characteristics of AZO/Ag/AZO. (**a**) Schematic of the sputtering of the AZO–Ag–AZO multilayer. (**b**) The simulated transmittance distribution of the AZO/Ag/AZO according to the Ag thickness and the AZO thickness in the AZO/Ag/AZO multilayer. (**c**) The simulated transmittance of the AZO–Ag–AZO according to the wavelength for different AZO thickness in the AZO/Ag/AZO multilayer with fixed Ag thickness of 12 nm. (**d**) The measured transmittance of the AZO–Ag–AZO according to the wavelength for different Ag thickness in the AZO/Ag/AZO multilayer with fixed AZO thickness of 32 nm. The measured transmittance of 12 nm Ag without AZO was also obtained and is shown in the graph. (**e**) Sheet resistance of AZO–Ag–AZO according to the Ag thickness in (**d**). (**f**) FOM value of the AZO–Ag–AZO at a wavelength of 550 nm, according to the Ag thickness in (**d**).

**Figure 2 micromachines-15-00146-f002:**
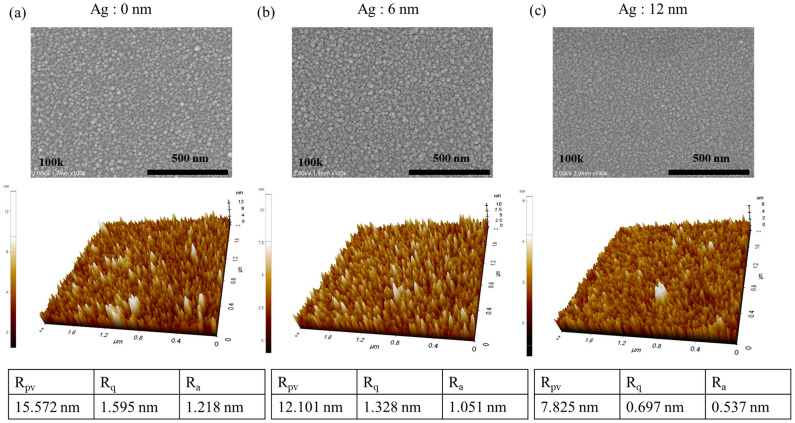
SEM and AFM analysis of the AZO–Ag–AZO according to the Ag thickness in the AZO–Ag–AZO multilayer. (**a**) Surface morphology and roughness of the AZO–Ag–AZO multilayer without the Ag layer. (**b**) Surface morphology and roughness of the AZO–Ag–AZO multilayer with a Ag thickness of 6 nm. (**c**) Surface morphology and roughness of the AZO–Ag–AZO multilayer with a Ag thickness of 12 nm.

**Figure 3 micromachines-15-00146-f003:**
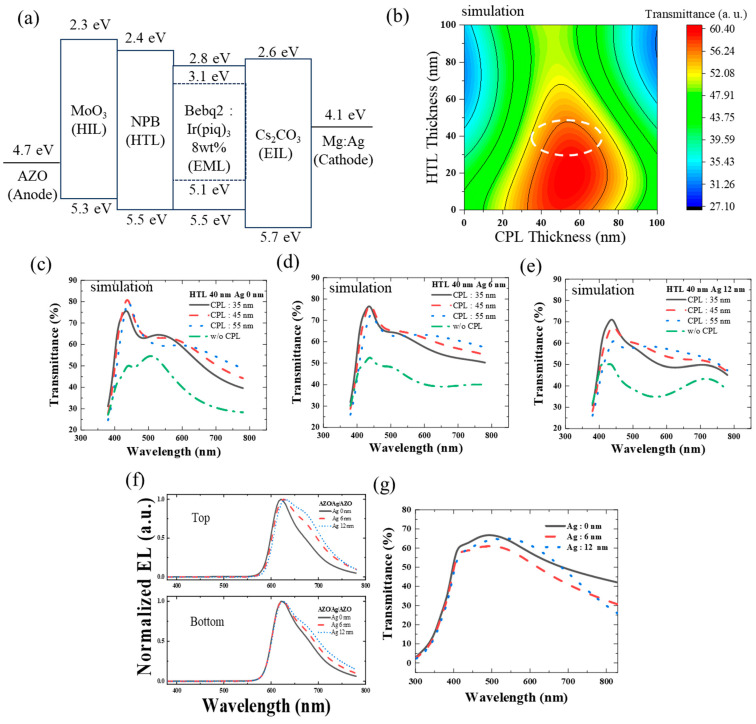
Concept of the optimized design of the transparent OLED with an AZO/Ag/AZO anode. (**a**) Schematic of the estimated energy band diagram of the red OLED (AZO/MoO_3_/NPB/Bebq_2_:Ir(piq)_3_ 8 wt.%/Cs_2_CO_3_/Mg:Ag). (**b**) The simulated transmittance distribution of the red OLED according to the HTL thickness and CPL thickness. (**c**) The simulated transmittance of the red OLED according to the wavelength for different CPL thickness without Ag layer in the AZO/Ag/AZO anode at a fixed HTL thickness of 40 nm. (**d**) The simulated transmittance of the red OLED according to the wavelength for different CPL thickness with a Ag thickness of 6 nm in the AZO/Ag/AZO anode at a fixed HTL thickness of 40 nm. (**e**) The simulated transmittance of the red OLED according to the wavelength for different CPL thickness with a Ag thickness of 12 nm in the AZO/Ag/AZO anode at a fixed HTL thickness of 40 nm. (**f**) The simulated normalized top and bottom EL of the red OLED according to the wavelength for different Ag thickness in the AZO/Ag/AZO anode with a CPL thickness of 45 nm. (**g**) The measured transmittance of the red OLED according to the wavelength for different Ag thickness in the AZO/Ag/AZO anode with a CPL thickness of 45 nm.

**Figure 4 micromachines-15-00146-f004:**
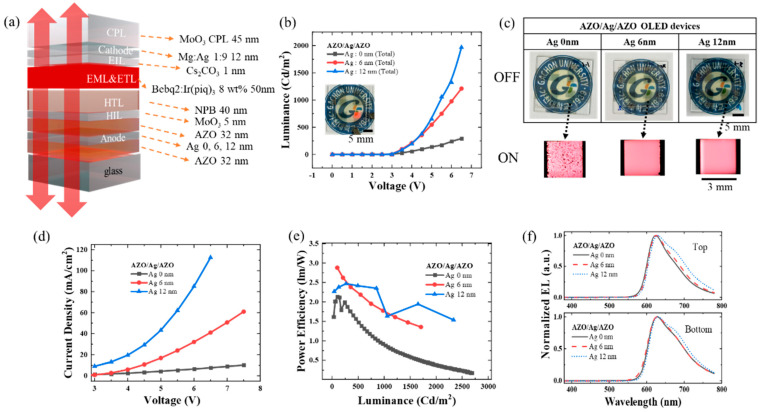
Analysis of the fabricated transparent OLED with different Ag thickness in the AZO/Ag/AZO anode. (**a**) Schematic of the fabricated red OLED (AZO/Ag/AZO anode/MoO_3_ HIL/NPB HTL/Bebq_2_: Ir(piq)_3_ 8 wt.% EML and ETL/Cs_2_CO_3_ EIL/Mg:Ag cathode/MoO_3_ CPL). (**b**) The luminance–voltage characteristics of the fabricated red OLED for different Ag thickness in the AZO–Ag–AZO anode. (**c**) The optical image of the transparent red OLED whether the voltage is applied (ON) or not (OFF) for different Ag thickness in the AZO/Ag/AZO anode. (**d**) The power efficiency–luminance characteristics of the fabricated red OLED for different Ag thickness in the AZO–Ag–AZO anode. (**e**) The current density–voltage characteristics of the fabricated red OLED for different Ag thickness in the AZO/Ag/AZO anode. (**f**) The normalized top and bottom EL of the fabricated red OLED according to the wavelength for different Ag thickness in the AZO/Ag/AZO anode.

**Table 1 micromachines-15-00146-t001:** Simulated and measured average transmittance of red OLED for a range of wavelengths from (380 to 780) nm according to various CPL thicknesses with different Ag thickness in the AZO/Ag/AZO anode.

	AZO/Ag/AZO 0 nm Ag	AZO/Ag/AZO 6 nm Ag	AZO/Ag/AZO 12 nm Ag
w/o CPL (simulation)	40.36	43.01	40.52
CPL 35 nm (simulation)	56.05	58.63	53.18
CPL 45 nm (simulation)	58.30	60.86	54.66
CPL 55 nm (simulation)	57.98	60.75	54.28
CPL 45 nm (Measured)	56.19	50.23	54.51

## Data Availability

The data supporting the findings of this study are available from the corresponding author upon request.

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
