# Peer review of "Highly Transparent Red Organic Light-Emitting Diodes with AZO/Ag/AZO Multilayer Electrode"

_micromachines, 2024, doi:10.3390/mi15010146_

Round 1

Reviewer 1 Report

Comments and Suggestions for Authors

In this manuscript, Lee et al. systematically investigated the transmittance and conductivity properties of AZO/Ag/AZO electrodes with varying thicknesses of Ag and their application as a transparent anode in transparent OLEDs. This work would be beneficial to researchers interested in conductive transparent electrodes. However, before recommending it for publication, the following issues should be addressed:

1. In the abstract, please provide the full names of FOM and TrOLED.

2. In comparison with the pure 12 nm Ag film, the AZO/Ag/AZO electrode with a 12 nm Ag layer exhibits a higher sheet resistance. In the measurement of sheet resistance, the Ag layer is in parallel with the two AZO layers in the AZO/Ag/AZO electrode. Therefore, it is reasonable to expect that the AZO/Ag/AZO electrode should have a lower sheet resistance. Could the authors provide an explanation for why the AZO/Ag/AZO electrode with a 12 nm Ag layer shows a higher sheet resistance?

3. In comparison with the pure 12 nm Ag film, the AZO/Ag/AZO electrode with a 12 nm Ag layer demonstrates higher transmittance. Please provide a reasonable explanation to clarify this discrepancy.

4. The transmittance characteristics of Mg:Ag, used as a semi-transparent cathode in transparent OLEDs, should be provided.

5. As mentioned by the authors, flexibility is a key factor for conductive transparent electrodes. Please provide the sheet resistance of the proposed electrode after bending to clarify its flexibility. Additionally, prepare and investigate a flexible transparent OLED with the proposed electrode.

6. Some references about flexible OLEDs (J. Mater. Chem. C 2014, 2, 835-840; Nat. Commun. 2023, 14, 1257; Light Sci. Appl. 2022, 11, 59) are missing. Please include these references in the manuscript.

Author Response

Responses to the reviewer 1’s comments

Reviewer #1 : In this manuscript, Lee et al. systematically investigated the transmittance and conductivity properties of AZO/Ag/AZO electrodes with varying thicknesses of Ag and their application as a transparent anode in transparent OLEDs. This work would be beneficial to researchers interested in conductive transparent electrodes. However, before recommending it for publication, the following issues should be addressed:

RESPONSE

We appreciate the reviewer’s comments on the validity and uniqueness of our study. The followings are responses to the individual comments.

Reviewer’s comment 1) In the abstract, please provide the full names of FOM and TrOLED.

RESPONSE

We appreciate the reviewer’s comments for bringing to us attention that we forgot to include full name. As suggested by the reviewers, we have added full names of FOM and TrOLED in the abstract.

Modification to the manuscript:

  • On line 19 ~21 in the manuscript

The AZO/Ag/AZO with Ag of 12 nm and AZO of 32 nm thickness achieved the results of the highest figure of merit (FOM) (Φ550 = 4.65 mΩ−1) and lowest roughness. The full structure of transparent OLED (TrOLED) with AZO/Ag/AZO anode and Mg:Ag cathode reached 64.84 % transmittance at 550 nm, and 300 cd/m2 at about 4 V.

Reviewer’s comment 2) In comparison with the pure 12 nm Ag film, the AZO/Ag/AZO electrode with a 12 nm Ag layer exhibits a higher sheet resistance. In the measurement of sheet resistance, the Ag layer is in parallel with the two AZO layers in the AZO/Ag/AZO electrode. Therefore, it is reasonable to expect that the AZO/Ag/AZO electrode should have a lower sheet resistance. Could the authors provide an explanation for why the AZO/Ag/AZO electrode with a 12 nm Ag layer shows a higher sheet resistance?

RESPONSE

We appreciate the reviewer’s comments of different sheet resistance of pure 12 nm Ag film and AZO/Ag/AZO electrode with a 12 nm Ag layer. We also agree that the resistance of both electrodes should be almost equal. It seems there might be a slight reproducibility error in our sputtering process. A re comparison was conducted by redepositing a pure 12 nm Ag film and AZO/Ag/AZO electrode with a 12 nm Ag layer. The purpose was to assess and analyze the differences between the two electrodes in terms of sheet resistance. The average sheet resistance of pure 12 nm Ag film was 12.49 Ω/sq with standard deviation 1.78 Ω/sq. The average sheet resistance of AZO/Ag/AZO electrode with a 12 nm Ag layer was 12.22 Ω/sq with standard deviation 0.268 Ω/sq. While samples deposited simultaneously exhibit the anticipated uniform resistance, minor variations with errors in reproducibility are might be happened in devices deposited over time. As suggested in reviewer comments, the sheet resistance graph in Figure 1e has been modified.

Modification to the Manuscript:

  • On line 178 in Manuscript

The single layer of 12 nm Ag condition (12.49 Ω/□) was the lowest condition, while the AZO/Ag/AZO multilayer with 12 nm Ag shows similar sheet resistance (12.22 Ω/□).

Reviewer’s comment 3) In comparison with the pure 12 nm Ag film, the AZO/Ag/AZO electrode with a 12 nm Ag layer demonstrates higher transmittance. Please provide a reasonable explanation to clarify this discrepancy.

RESPONSE

We appreciate the reviewer’s comments on how the the AZO/Ag/AZO electrode with a 12 nm Ag layer has higher transparency. The cause is clearly elucidated in the following paper. [Advanced Optical Materials 2021, 9, 2001298]. “The significantly improved transmittance over the visible range is associated with a substantially reduced reflection (from 38.3% to 6.0% by adding the two dielectric layers), which is caused by multiple optical resonances inside the two dielectric layers. Such optical resonance occurs at a certain wavelength when the associated net phase shift inside the dielectric layer equals to multiples of 2π radians. The net phase shift inside each dielectric layer includes both reflection phase shifts at the two metal– dielectric interfaces and the propagation phase shift through the dielectric layer.”

As suggested in reviewer comments, an explanation has been added to the text as to why the transmittance of AZO-Ag-AZO is higher than that of pure Ag film.

Modification to the Manuscript:

  • On line 175 in Manuscript

This is because optical resonance caused by two AZO film layers allows higher transmittance than pure Ag film [21].

Modification to the Manuscript:

  • On line 464 in Manuscript

  1. Zhang, C.; Ji, C.; Park, Y.-B.; Guo, L. J. Thin-Metal-Film Based Transparent Conductors: Material Preparation, Optical Design, and Device Applications. Advanced Optical Materials 2021, 9, 2001298, doi:10.1002/adom.202001298.

Reviewer’s comment 4) The transmittance characteristics of Mg:Ag, used as a semi-transparent cathode in transparent OLEDs, should be provided.

RESPONSE

 We appreciate the reviewer’s comments on semi-transparent cathode which is very important at transparent OLEDs. Through previous research, we reported a paper on transparency analysis according to the Mg:Ag ratio [Materials Chemistry and Physics 2023, 303, 127742.]. We cited the relevant prior research and added related explanations to the text.

Modification to the Manuscript:

  • On line 270 in Manuscript

The transmittance of 12nm Mg:Ag deposited at 1:9 ratio was 63.69% [26].

Modification to the Manuscript:

  • On line 476 in Manuscript

  1. Lee, D.; Cho, E.-S.; Jeon, Y.; Kwon, S.J. Characterization of the material and electrical properties depending on the Mg: Ag ratio as a cathode for TEOLED. Materials Chemistry and Physics 2023, 303, 127742.

Reviewer’s comment 5) As mentioned by the authors, flexibility is a key factor for conductive transparent electrodes. Please provide the sheet resistance of the proposed electrode after bending to clarify its flexibility. Additionally, prepare and investigate a flexible transparent OLED with the proposed electrode.

RESPONSE

We appreciate the reviewer’s comments on bending electrode to clarify its flexibility. Many previous studies have been reported on the flexibility analysis of AZO/Ag/AZO. For example, with AZO/Ag/AZO with flexible substrate, the resistance remained stable without increase until it was bent to 149 degrees. [Ceramics International 2021, 47, 5671-5676].

 Unlike ITO, AZO/Ag/AZO electrode exhibited no change in resistance and remained stable during 100 consecutive bending cycles. [Solar Energy Materials and Solar Cells 2017, 165, 88-93].

The D/M/D structure, like AZO/Ag/AZO, exhibits good flexibility when subjected to bending.[ J. Mater. Chem. C 2014, 2, 835-840]. And it can be seen that the bending characteristics of the AZO thin film are excellent. [Organic Electronics 2013, 14, 236-249]. And the possibility of bending of OLED with AZO electrode was also reported.[Journal of Physics D: Applied Physics 2010, 43, 465403]. Based on the reviewer's comments, a discussion on the flexibility of AZO-Ag-AZO electrodes and their applicability to flexible OLEDs was added to the text.

Modification to the Manuscript:

  • On line 376 in Manuscript

The successful result without thermal treatment for the AZO/Ag/AZO TCE and TrOLED also indicates the possibility of adopting flexible substrates with low thermal stability, such as PET [40-44].

Modification to the Manuscript:

  • On line 512 in Manuscript

  1. Liu, X.-N.; Gao, J.; Gong, J.-H.; Wang, W.-X.; Chen, S.-C.; Dai, M.-J.; Lin, S.-S.; Shi, Q.; Sun, H. Optoelectronic properties of an AZO/Ag multilayer employed as a flexible electrode. Ceramics International 2021, 47, 5671-5676, doi:https://doi.org/10.1016/j.ceramint.2020.10.153.
  2. Torrisi, G.; Crupi, I.; Mirabella, S.; Terrasi, A. Robustness and electrical reliability of AZO/Ag/AZO thin film after bending stress. Solar Energy Materials and Solar Cells 2017, 165, 88-93, doi:https://doi.org/10.1016/j.solmat.2017.02.037.
  3. Liu, S.; Liu, W.; Yu, J.; Zhang, W.; Zhang, L.; Wen, X.; Yin, Y.; Xie, W. Silver/germanium/silver: an effective transparent electrode for flexible organic light-emitting devices. J. Mater. Chem. C 2014, 2, 835-840, doi:10.1039/c3tc31927j.
  4. Lei, P.-H.; Hsu, C.-M.; Fan, Y.-S. Flexible organic light-emitting diodes on a polyestersulfone (PES) substrate using Al-doped ZnO anode grown by dual-plasma-enhanced metalorganic deposition system. Organic Electronics 2013, 14, 236-249, doi:https://doi.org/10.1016/j.orgel.2012.10.030.
  5. Jeong, J.-A.; Shin, H.-S.; Choi, K.-H.; Kim, H.-K. Flexible Al-doped ZnO films grown on PET substrates using linear fac-ing target sputtering for flexible OLEDs. Journal of Physics D: Applied Physics 2010, 43, 465403, doi:10.1088/0022-3727/43/46/465403.

Reviewer’s comment 5) Some references about flexible OLEDs (J. Mater. Chem. C 2014, 2, 835-840; Nat. Commun. 2023, 14, 1257; Light Sci. Appl. 2022, 11, 59) are missing. Please include these references in the manuscript.

RESPONSE

We appreciate the reviewer's comments on references of flexible OLEDs. We included those references.

Modification to the Manuscript:

  • On line 76 in Manuscript

A transparent organic light-emitting diode (OLED) display is a highly promising de-vice for the next-generation displays, including AR/VR/MR, smart glass, and smart windows, due to its advantageous characteristics of flexibility [12-14], and self-emissive properties with optical clarity.

Modification to the Manuscript:

  • On line 446 in Manuscript

  1. Liu, S.; Liu, W.; Yu, J.; Zhang, W.; Zhang, L.; Wen, X.; Yin, Y.; Xie, W. Silver/germanium/silver: an effective transparent electrode for flexible organic light-emitting devices. J. Mater. Chem. C 2014, 2, 835-840, doi:10.1039/c3tc31927j.
  2. Zhang, Q.; Xu, M.; Zhou, L.; Liu, S.; Wang, W.; Zhang, L.; Xie, W.; Yu, C. A flexible organic mechanoluminophore device. Nature Communications 2023, 14, doi:10.1038/s41467-023-36916-z.
  3. Pan, T.; Liu, S.; Zhang, L.; Xie, W.; Yu, C. A flexible, multifunctional, optoelectronic anticounterfeiting device from high-performance organic light-emitting paper. Light: Science & Applications 2022, 11, 59, doi:10.1038/s41377-022-00760-5.

Again, we would like to express our appreciation for the reviewers’ thoughtful suggestions. We have been able to revise and improve the paper as a result of the reviewers’ valuable feedback.

We look forward to your positive response. Thank you very much for your time and attention.

Sincerely yours,

Yongmin Jeon and Eou-Sik Cho

Professor

Gachon University

Seongnam 13120, Republic of Korea.

Phone: +82-31-750-2618

E-mail: yongmin@gachon.ac.kr, es.cho@gachon.ac.kr

Reviewer 2 Report

Comments and Suggestions for Authors

This paper reports some original and very interesting results on AZO/Ag/AZO multilayer as transparent Electrode used in organic light-emitting diodes. The topic is of high importance in the optoelectronic field and the AZO/Ag/AZO systems had never been explored in this target and details. This paper is interesting because it completes an important part of the photophysics of this multilayers systems.

 This paper is clear and well organized and I do not find any flaws in this work. The synthesis and experimental spectra are well illustrated and easily to understand, the interpretation and conclusion are convincing. Finally, this work is of great interest; to my mind, it is suitable for publication.

 Minor revision: 

To open this article to a wider audience, you should avoid putting too many abbreviations in the abstract. I suggest to complete in the abstract “ TrOLED" and "FOM" for instance.

Author Response

Responses to the reviewer 2’s comments

Reviewer #2 : This paper reports some original and very interesting results on AZO/Ag/AZO multilayer as transparent Electrode used in organic light-emitting diodes. The topic is of high importance in the optoelectronic field and the AZO/Ag/AZO systems had never been explored in this target and details. This paper is interesting because it completes an important part of the photophysics of this multilayers systems.

 This paper is clear and well organized and I do not find any flaws in this work. The synthesis and experimental spectra are well illustrated and easily to understand, the interpretation and conclusion are convincing. Finally, this work is of great interest; to my mind, it is suitable for publication.

RESPONSE

We thank the reviewer for the fruitful comments on our work. We agree with the reviewers and have revised the manuscript to help readers better understand the paper. The followings are responses to the individual comments.

Reviewer’s comment 1) To open this article to a wider audience, you should avoid putting too many abbreviations in the abstract. I suggest to complete in the abstract “ TrOLED" and "FOM" for instance

RESPONSE

We appreciate the reviewer’s comments for bringing to us attention that we forgot to include full name. As suggested by the reviewers, we have added full names of FOM and TrOLED in the abstract.

Modification to the manuscript:

  • On line 19 ~21 in the manuscript

The AZO/Ag/AZO with Ag of 12 nm and AZO of 32 nm thickness achieved the results of the highest figure of merit (FOM) (Φ550 = 4.65 mΩ−1) and lowest roughness. The full structure of transparent OLED (TrOLED) with AZO/Ag/AZO anode and Mg:Ag cathode reached 64.84 % transmittance at 550 nm, and 300 cd/m2 at about 4 V.

Again, we would like to express our appreciation for the reviewers’ thoughtful suggestions. We have been able to revise and improve the paper as a result of the reviewers’ valuable feedback.

We look forward to your positive response. Thank you very much for your time and attention.

Sincerely yours,

Yongmin Jeon and Eou-Sik Cho

Professor

Gachon University

Seongnam 13120, Republic of Korea.

Phone: +82-31-750-2618

E-mail: yongmin@gachon.ac.kr, es.cho@gachon.ac.kr

Reviewer 3 Report

Comments and Suggestions for Authors

1)    The authors have shown in Line 248 that for all conditions of Ag thicknesses for AZO/Ag/AZO anode, their simulated result shows more transparency than that without CPL. Typically, when we add on more layers, the transmittance will decrease unless it has anti-reflection effect. However, adding the AZO/Ag/AZO layer, the transmittance has increased for all Ag thicknesses. Do the authors have any explanation of why without CPL has lower transmittance?

In addition to this main comment,

In this manuscript, the reviewer would like to add: The authors are using a newer type of TCO/M/TCO electrode on OLED for future flexible transparent device. They have detailed their TCO/M/TCO structure from simulation and tally with experiment. Then, they even demonstrate their new TCO stack on a working OLED and compare their efficiency of each TCO/M/TCO stack.

The reviewer feels the whole manuscript is relevant to the field of flexible OLED. It is addressing on a newer TCO/M/TCO stack as electrode for use in flexible OLED. Though there are not the first one to report this kind of TCO/M/TCO stack, it is still meaningful to examine the optimized thickness of the TCO/M/TCO stack and apply it on working OLED.

The novelty of this publication is that other publications only demonstrate the TCO/M/TCO by pure experiment, not every publications include a simulation result and a working application.

What specific improvements should the authors consider regarding the methodology? What further controls should be considered? No comments. Already a well-detailed experiment done on glass. Just that it is not yet apply onto real flexible device.

Please describe how the conclusions are or are not consistent with the evidence and arguments presented. Please also indicate if all main questions posed were addressed and by which specific experiments. The conclusion is consistent with the results discussion. The authors have stated that their process is fully room temperature and their TCO/M/TCO stack has very low roughness which is suitable for future flexible device. 

Please note that there are some typos in the manuscript.

Comments on the Quality of English Language

Minor comments:

1)    The authors have used many abbreviations in the manuscript. It would be great if all the abbreviations have been first represented before they are used frequently in the subsequent paragraphs. This will help readers who are new to this research understand better.

2)    In the manuscript, the authors have 2 conclusions: one at the end of section 3: results and discussions and one at the section 4 conclusion. It would be great to rephrase to have one single conclusion section.

 3)    Observe that they are some typos in the manuscript.

Figure 3(b) Transmittancce?

Figure S1. ag12 nm.

Figure S2. Captions. Thickess vs thicknesses

Figure S5. Figure captions of the figures are wrong. (a) should be (b). (b) should be (a)

Author Response

Responses to the reviewer 3’s comments

Reviewer #3 : In this manuscript, the reviewer would like to add: The authors are using a newer type of TCO/M/TCO electrode on OLED for future flexible transparent device. They have detailed their TCO/M/TCO structure from simulation and tally with experiment. Then, they even demonstrate their new TCO stack on a working OLED and compare their efficiency of each TCO/M/TCO stack.

The reviewer feels the whole manuscript is relevant to the field of flexible OLED. It is addressing on a newer TCO/M/TCO stack as electrode for use in flexible OLED. Though there are not the first one to report this kind of TCO/M/TCO stack, it is still meaningful to examine the optimized thickness of the TCO/M/TCO stack and apply it on working OLED.

The novelty of this publication is that other publications only demonstrate the TCO/M/TCO by pure experiment, not every publications include a simulation result and a working application.

RESPONSE

Thank you for reviewing our research and recognizing its significance and novelty. The followings are responses to the individual comments.

Reviewer’s comment 1) The authors have used many abbreviations in the manuscript. It would be great if all the abbreviations have been first represented before they are used frequently in the subsequent paragraphs. This will help readers who are new to this research understand better.

RESPONSE

We appreciate the reviewer’s comments for bringing to us attention that we forgot to include full name. As suggested by the reviewers, we have added full names.

Modification to the manuscript:

  • On line 19 ~21 in the manuscript

The AZO/Ag/AZO with Ag of 12 nm and AZO of 32 nm thickness achieved the results of the highest figure of merit (FOM) (Φ550 = 4.65 mΩ−1) and lowest roughness. The full structure of transparent OLED (TrOLED) with AZO/Ag/AZO anode and Mg:Ag cathode reached 64.84 % transmittance at 550 nm, and 300 cd/m2 at about 4 V.

Modification to the manuscript:

  • On line 43 in the manuscript

This is becoming increasingly evident in the development of the next generation dis-plays, such as light emitting diodes (LED) and OLEDs, among others [5].

Modification to the manuscript:

  • On line 50 in the manuscript

real-time color changing of the OLED synchronized with the shape of the measured (top) normal electrocardiography (ECG) signal and (bottom) abnormal ECG signal

Reviewer’s comment 2) In the manuscript, the authors have 2 conclusions: one at the end of section 3: results and discussions and one at the section 4 conclusion. It would be great to rephrase to have one single conclusion section.

RESPONSE

We appreciate the reviewer's comments regarding the suggestion to rephrase and consolidate into a single conclusion. We totally agree and rephrased to one single conclusion.

Modification to the manuscript:

  • On line 336 ~345 in the manuscript

we successfully implemented the AZO/Ag/AZO TCE for the TrOLED anode.

In conclusion, we successfully implemented the AZO/Ag/AZO TCE for the TrOLED anode. The optimum condition of the AZO/Ag/AZO with thickness of 32 nm/12 nm/32 nm demonstrated the best results when considering the FOM, surface roughness, and overall device transmittance in the previous analysis, and also, the best TrOLED characteristics. To the best of our knowledge, this work is the world’s first TrOLED, and is suitable for applications such as AR, smart glasses and MR, with the AZO/Ag/AZO electrode that is capable of maximizing the transparency and flexibility for the wearable device. Furthermore, compared to other studies where the AZO or AZO/Ag/AZO anode is used for OLED, despite being a TrOLED, the characteristics of our device are impressive [26−34].

 Modification to the manuscript:

  • On line 360 in the manuscript

In conclusion, the AZO/Ag/AZO multilayers are

Modification to the manuscript:

  • On line 371 in the manuscript

To the best of our knowledge, this work is the world’s first TrOLED, and is suitable for applications such as AR, smart glasses, and MR, with the AZO/Ag/AZO electrode that is capable of maximizing the transparency and flexibility for the wearable device. Furthermore, compared to other studies where the AZO or AZO/Ag/AZO anode is used for OLED, despite being a TrOLED, the characteristics of our device are impressive [30−38].

Modification to the manuscript:

  • On line 379 in the manuscript

for the next generation display and We hope this work not limited to OLED’s electrode.

Reviewer’s comment 3) Observe that they are some typos in the manuscript.

RESPONSE

We appreciate the reviewer’s comments for bringing to us attention to our mistakes. It was corrected to address the errors.

  • Figure 3(b) Transmittancce

Modification to the manuscript:

-            On page 8 in the manuscript

  • Figure S1. ag12 nm.

Modification to the Supporting Information:

  • On page 2 in Supporting Information
  • Figure S2. Captions. Thickess vs thicknesses

Modification to the Supporting Information:

  • On page 3 in Supporting Information

thickness

  • Figure S5. Figure captions of the figures are wrong. (a) should be (b). (b) should be (a)

Modification to the Supporting Information:

  • On page 6 in Supporting Information t

(a) the current density – voltage characteristics of the fabricated red OLED for different Ag thickness in the AZO/Ag/AZO anode; (b) The luminance–voltage characteristics of the fabricated red OLED for different Ag thickness in the AZO/Ag/AZO anode;

Again, we would like to express our appreciation for the reviewers’ thoughtful suggestions. We have been able to revise and improve the paper as a result of the reviewers’ valuable feedback.

We look forward to your positive response. Thank you very much for your time and attention.

Sincerely yours,

Yongmin Jeon and Eou-Sik Cho

Professor

Gachon University

Seongnam 13120, Republic of Korea.

Phone: +82-31-750-2618

E-mail: yongmin@gachon.ac.kr, es.cho@gachon.ac.kr
